# Aeolian sediment transport on Io from lava–frost interactions

George D. McDonald [1✉], Joshua Méndez Harper[2], Lujendra Ojha[1], Paul Corlies[3], Josef Dufek[2], Ryan C. Ewing [4] & Laura Kerber[5]

Surface modification on Jupiter's volcanically active moon, Io, has to date been attributed almost exclusively to lava emplacement and volcanic plume deposits. Here we demonstrate that wind-blown transport of sediment may also be altering the Ionian surface. Specifically, shallow subsurface interactions between lava and Io's widespread sulfur dioxide ($SO_2$) frost can produce localized sublimation vapor flows with sufficient gas densities to enable particle saltation. We calculate anticipated outgassing velocities from lava–$SO_2$ frost interactions, and compare these to the saltation thresholds predicted when accounting for the tenuous nature of the sublimated vapor. We find that saltation may occur if frost temperatures surpass 155 K. Finally we make the first measurements of the dimensions of linear features in images from the *Galileo* probe, previously termed "ridges", which demonstrate certain similarities to dunes on other planetary bodies. Io joins a growing list of bodies with tenuous and transient atmospheres where aeolian sediment transport may be an important control on the landscape.

[1] Department of Earth and Planetary Sciences, Rutgers, The State University of New Jersey, Piscataway, NJ, USA. [2] Department of Earth Sciences, University of Oregon, Eugene, OR, USA. [3] Department of Earth, Atmospheric and Planetary Sciences, Massachusetts Institute of Technology, Cambridge, MA, USA. [4] Department of Geology and Geophysics, Texas A&M University, College Station, TX, USA. [5] Jet Propulsion Laboratory, California Institute of Technology, Pasadena, CA, USA. ✉email: gm686@rutgers.edu

Jupiter's moon Io is the most volcanically active body in the solar system, and the body that is known to be most rapidly resurfaced[1]. The vigor with which new material is deposited from lava flows and plumes has meant that Io's surface modification has been studied almost exclusively with respect to this initial emplacement of material. Nevertheless, this same dynamic environment, where hot, recently erupted material is constantly interacting with extensive volatiles on the surface, may enable a host of secondary surface modification mechanisms.

Saltation is the process by which near-surface turbulence in the carrier fluid mobilizes and lifts grains, followed by ballistic grain trajectories that lead to splashing and the ejection of additional grains[2]. The transport of sediment via saltation gives rise to macroscale aeolian bedforms, including ripples, dunes and via erosion, yardangs.

Io possesses an atmosphere which could potentially act as a carrier fluid for saltation. This atmosphere exhibits significant variability, with order of magnitude decreases in surface pressure poleward of 45° latitude, up to order $10^1$ longitudinal variations, and order $10^{-1}$ enhancements over volcanic plumes at the ~100 km scale[3,4]. This variability drives ~300 m/s winds, which are thought to primarily be a day-to-night circulation[5]. Regional flows from areas of higher atmospheric density in the vicinity of volcanic vents to surrounding terrain may be superposed over the prevailing diurnal circulation[6–8]. Despite the variability, this atmosphere is collisionally thick and permanently detectable. However, with peak surface pressures of ~1 nbar, the tenuous nature of this ambient atmosphere is perhaps the greatest impediment to saltation on Io.

Previous studies identified linear features occurring on otherwise flat terrain, which they termed "ridges", and noted the morphological similarity of these features to dunes on Earth and Mars[9,10]. However, choosing 0.1 nbar as a representative atmospheric pressure, one study estimated a 20 km/s threshold friction speed to move grains on Io[10], two orders of magnitude greater than Io's ~300 m/s winds[7]. Concluding that an aeolian origin for these features was impossible, formation by tidal forces was favored[10]. More recent studies have concurred that while saltation under Io's ambient atmosphere seems unlikely, there could be locally substantially thicker portions of the atmosphere in the vicinity of volcanic vents, and that an aeolian origin for the ridges cannot be entirely ruled out[11]. The ridges' regular spacing, slightly meandering form, and possession of crestline defects have been mentioned as dune-like characteristics[12].

Here, we demonstrate that interactions between lava from volcanic eruptions and the sulfur dioxide ($SO_2$) frost blanketing Io's surface can produce localized sublimation vapor flows with sufficient gas densities to enable saltation. We first review and confirm that saltation under Io's ambient atmosphere seems implausible. Then, we demonstrate how basaltic lava flows beneath snowpacks on Earth suggest a mechanism for raising the sublimation vapor pressure of $SO_2$ above ambient atmospheric pressure on Io. We calculate anticipated outgassing velocities from lava-$SO_2$ frost interactions, and compare these to the saltation thresholds predicted when accounting for the tenuous nature of the sublimated vapor. Finally, we turn to quantifying observational evidence for aeolian transport on Io. We make the first measurements of the dimensions of the Ionian ridges, including their crestline defect density to crest spacing ratio, which can be directly compared with dunes on other planetary bodies. We also estimate the topographic profile of the ridges using a rudimentary photoclinometric analysis.

## Results

### Lava–frost interactions and sublimation vapor pressures.
On Earth, winds are effectively always turbulent at length scales relevant to the saltation of sand dune grains. For this reason, grain scale turbulence is often not explicitly considered as a precondition for saltation. Nevertheless it is the vertical fluid advection from turbulent eddies which is required to lift, and thereby saltate, particles[2]. While laminar flows possess vertical shear which can initiate particle motion, under these conditions sediment transport manifests solely as particle creep—transport via saltation will not occur due to the lack of vertical fluid advection[13]. For conditions as tenuous as the Ionian atmosphere of ~1 nbar atmospheric pressure, it is not apparent at all whether flows at the grain scale would be turbulent. We therefore explicitly consider turbulent flow to be a precondition for saltation. In assessing turbulence relevant to the grain scale, we consider the Reynolds number (Re) at the viscous sublayer that forms within the atmosphere in immediate proximity to the surface (see Methods). The Reynolds number is a non-dimensional number that is used to assess whether a flow is laminar or turbulent. We describe flows with $Re > 10^3$ as transitionally turbulent.

Whether we consider a typical dayside atmospheric density of $4 \times 10^{-9}$ kg/m$^{3}$[3,4], or the atmosphere with an enhanced density of $7 \times 10^{-9}$ kg/m$^3$ in the immediate vicinity of volcanic vents[14], we calculate $10^{-5} < Re < 10^{-1}$. This indicates that the low density of Io's ambient atmosphere results in laminar flow, precluding saltation under these ambient conditions (see Fig. 1a; see Methods).

Nevertheless, effusive eruptions, which are typified by Prometheus Patera, provide a continuous and steady source of silicate lava onto the surface[15,16], and may provide a mechanism to generate turbulent gas flows at the Ionian surface. Paterae are the Ionian analogue to terrestrial calderas, or volcanic craters. These lavas can reach temperatures of 1041 K at the surface[17], which is roughly 900 K above the typical daytime surface temperatures of 107 K at Prometheus Patera[14]. Volcanic plumes, 100s of km in scale and typically originating from the primary volcanic vent, are also thought to the be the source of extensive volatiles. These are predominantly in the form of $SO_2$ frost, with $SO_2$-rich regions covering >60% of Io's surface[18]. At 107 K and $6 \times 10^{-10}$ bar, which is the pressure corresponding to the previously quoted typical atmospheric density, $SO_2$ in the solid phase is thermodynamically favored. With an isobaric slight increase in temperature to 110 K, the gas phase becomes favored (see arrow II in Fig. 2a). Sublimation at atmospheric pressure results in a vapor density that is too low for turbulent flow, given the ~300 m/s wind speeds on Io (i.e. $Re = 10^{-5}$. See Fig. 1a; see Methods). Nonetheless, any heat transfer from the magma to this $SO_2$ frost may provide a means to dramatically raise the sublimation vapor pressure by means of the Clausius-Clapeyron relation whereby $P_{sub} \propto \exp(-L/(RT_{sub}))$. Here, $P_{sub}$ and $T_{sub}$ are the constant pressure and temperature, respectively, during the solid to gas phase change, $L$ is the specific latent heat, and $R$ is the gas constant. For an increase in the sublimation vapor pressure, the elevation in temperature of the $SO_2$ would need to be accompanied by increases in the ambient pressure (see arrow I in Fig. 2a).

Terrestrial pahoehoe, effusive lava flows with smooth surfaces, in the presence of thick snowpack advance beneath or within the snow[19]. Analogous geometries could be expected for effusive flows on Io advancing into $SO_2$ frost. In this scenario, the basal pressure from the overlying solid $SO_2$ would raise the ambient pressure of the sub-surface region where lava is flowing into and sublimating the $SO_2$ (see Fig. 2b).

Modest solid $SO_2$ depths from 0.1 to 0.5 m would apply sufficient basal pressure ($3 \times 10^{-3}$ to $2 \times 10^{-2}$ bar, see shaded region of Fig. 2a) so as to reach the sublimation curve for any temperatures up to the $SO_2$ triple point of 197.64 K. Any thicker snowpack will also lead to $SO_2$ vapor production, but the

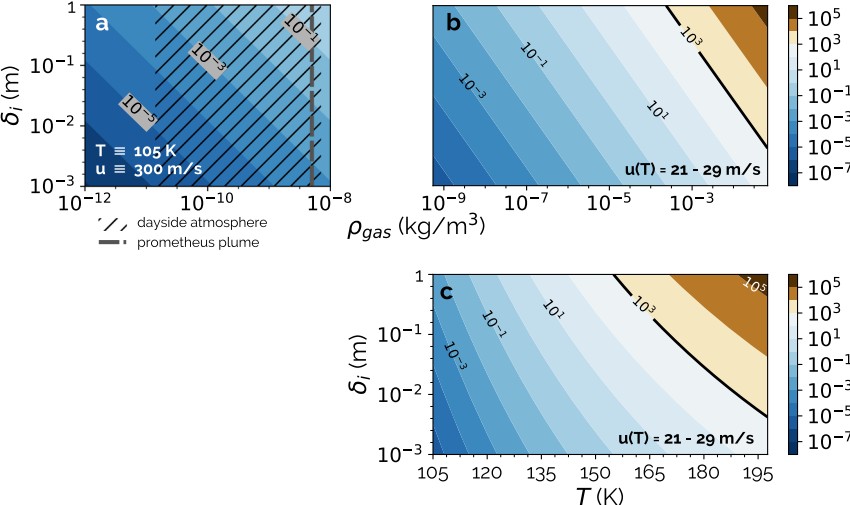

**Fig. 1 Reynolds number (Re) contours. a** For the atmospheric densities ($\rho_{gas}$) representative of Io's ambient atmosphere, with the hatching and the dashed line indicating parts of the parameter space occupied by specific geographical regions. The floating powers of 10 are labels for the contours. The constant temperature ($T$) and wind velocity ($u$) used for this calculation are denoted. Calculations in **a**, **b** and **c** are a function of the atmospheric viscous sublayer height $\delta_i$ which is not known for Io. **b** For the modeled sublimating vapor at the lava-$SO_2$ frost margins, as a function of the outflowing vapor density ($\rho_{gas}$). Note that in **b**, the calculated Re is a function of temperature, as the outflowing vapor density is dependent on temperature. The outgassing velocity, $u_T$, that Re is evaluated at in **b** and **c** is also a function of temperature. The contour for the critical Re value is denoted by the thick solid line. **c**, The same results shown in **b**, explicitly showing the dependence on $SO_2$ sublimation temperature.

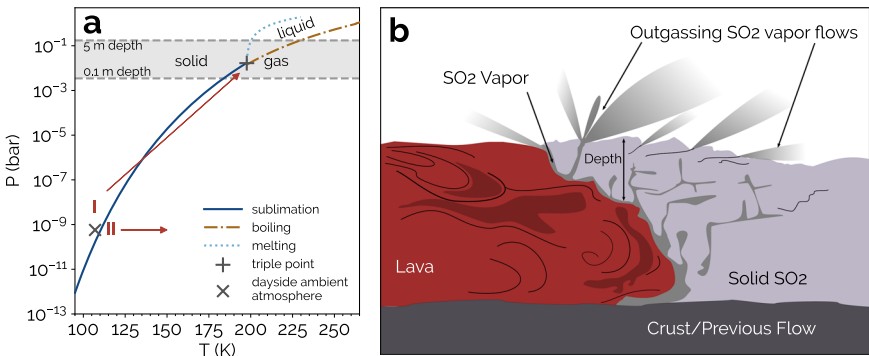

**Fig. 2 Above ambient sublimation vapor pressures on Io. a** The $SO_2$ phase diagram, with the phase transitions depicted as lines. The triple point and representative dayside ambient conditions are marked. The shaded box denotes basal pressures that could be exerted by solid $SO_2$ capping the lava-$SO_2$ interaction region. The arrows depict alternate thermodynamic pathways following the lava-frost interaction. **b** The geometry of the considered lava-$SO_2$ frost interaction. The light grey regions all consist of $SO_2$ vapor, which outgas as vapor flows upon reaching the surface.

process will be more complex and analogous to that on Earth where significant energy is expended in melting, prior to vapor production via boiling[20].

**Outgassing velocities and conditions for saltation.** We now consider the properties of $SO_2$ outgassing with vapor pressures of $2 \times 10^{-5}$ to $5 \times 10^{-3}$ bar, equivalent to the sublimation curve for temperatures of 145–185 K. We borrow from studies of comet outgassing, where the analogous sublimation of volatiles within a tenuous coma of densities $\sim 10^{-15} - 10^{-4}$ kg/m³ is well studied. On comets, the sublimating water vapor rapidly expands into an environment with orders of magnitude lower ambient pressure. A pure, sublimating volatile will outgas at the speed of sound[21,22]. While sonic outgassing is a possibility, whether saltation can occur in such a sonic flow is currently not well established[23]. We consider, however, that the $SO_2$ on Io is likely not sublimating directly into the ambient atmosphere. At the lava-frost margins, extensive granular material is likely present, both as a result of lava fragmentation[24] and the $SO_2$ frost being snow-like, or effectively granular, with inferred sizes of 50–250 μm in the

Prometheus Patera region[25]. For comet outgassing where the sublimating layer is capped by and sublimates through a layer of dust, relations have been derived indicating subsonic outgassing velocities[22]. Furthermore, dust-loading of the outgassing vapor can lead to steady-state subsonic velocities[21]. Applying the comet outgassing relations[22] for $SO_2$ outgassing at our considered vapor pressures, we find outgassing velocities of 21–29 m/s. The outgassing velocity is a function of the probability $p_c$ for a gas molecule to pass through the overlying porous layer, which is in turn a convolution of the height of the granular layer as well as its porosity. We find that varying $p_c$ between $0.01 < p_c < 0.5$ has a <2% effect on the calculated outgassing velocity, and thus for simplicity we assume going forward $p_c = 0.07$ as used by for Comet 67P/Churyumov-Gerasimenko[22] (hereafter, Comet 67P. See Supplementary Fig. 1; see Methods). For this value of $p_c$, all outgassing velocities correspond to a Mach value of 0.157—thus well-established saltation relations for subsonic conditions can be used.

With regards to the geometry of the flow, the outgassing vapor will act as a momentum driven plume, due to the overwhelming strength of the pressure gradient force in comparison with body

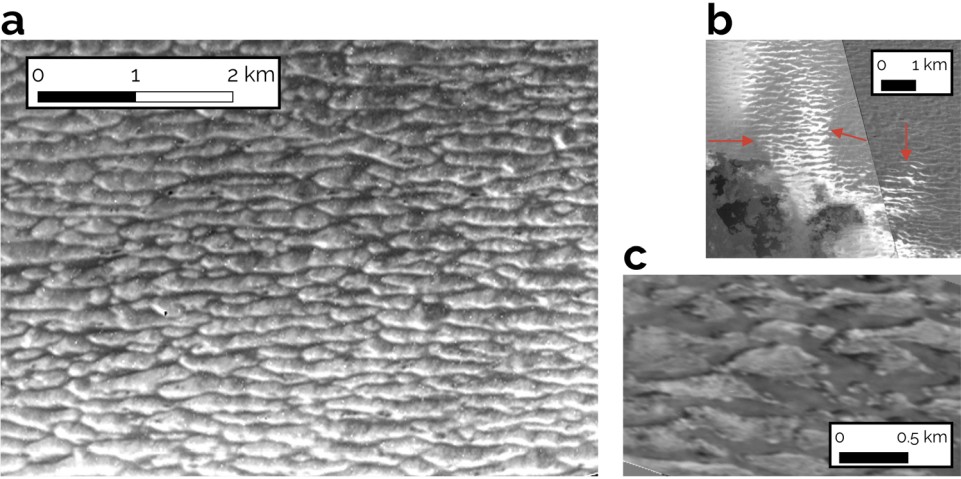

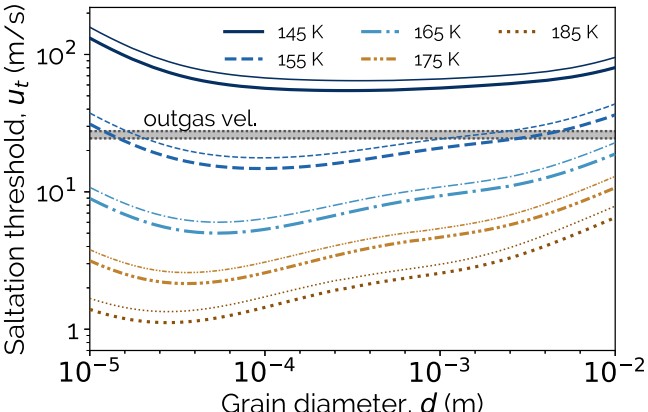

**Fig. 3 Ridges of possible aeolian origin from the _Galileo_ Solid State Imager. a** Adjacent to Prometheus Patera, within kilometers of the lava flow field. **b** Potential evidence of vapor flow indicated by the arrows at the margin of the same ridge field shown in **a**. **c** Ridges near Chaac Patera. Geographic coordinates and image IDs are tabulated in Supplementary Table 2.

forces such as gravity. Thus the direction of a given vapor flow will be influenced by the initial velocity of the lava at the lava–frost interfrace. While flows returning in the direction of the impinging lava may be possible given complex interaction geometries, it is expected that the majority of flows will be radial to the lava-frost interface plane, and thus outward with respect to the paterae where lava flows originate (grey regions labeled "outgassing vapor flows" in Fig. 2b).

High albedo streaks over the frost adjacent to lava flows at Prometheus Patera could be interpreted as evidence for this phenomenon (see Fig. 3b). It has been noted that the bright material only coats slopes facing the lava-flow, indicating radially outward deposition in low-angle jets[11,20].

It is this outgassing of $SO_2$ vapor that we suggest may transport grains. We first consider whether grains may be initially, directly entrained within the vapor flow. For this, we calculate the Stokes number (Stk), which is a nondimensional parameter that indicates the coupling of suspended particles to their carrier fluid. For Stk ≫ 1, particles are poorly coupled to their carrier fluid and largely follow ballistic trajectories. For our considered vapor pressures and particle diameters of 0.1 $\mu$m – 1 cm, we calculate Stokes numbers of $10^3 <$ Stk $< 10^{10}$ (see Supplementary Fig. 2; see Methods), suggesting that there will be little direct entrainment of granular material within the outgassing flow.

Nevertheless, we find that at the saturation vapor pressure, for temperatures T ≥ 155 K, the vapor flow can be turbulent, with Re > $10^3$ (see Fig. 1c). With the presence of turbulent flow, we can consider the transport of granular material via saltation, even if there is little initial entrainment via gas-grain coupling. Relations commonly used in planetary science for calculating the threshold shear velocity above which saltation is possible, assume a no-slip condition for the particle drag force[26,27]. The validity of the no-slip condition is a function of the Knudsen number (Kn), which is the ratio of the gas mean free path to the diameter of the grain under consideration. For Kn < 0.01, such as on Earth and for certain grain sizes on Mars, the no-slip condition is valid. Nevertheless, a large fraction of the parameter space that we consider for Io corresponds to Kn > 0.01 (see Methods; see Supplementary Fig. 3). We adopt the saltation relation derived for application to Comet 67P, which can incorporate modifications to the drag relations needed in the high Kn regime[22].

Figure 4 shows the saltation threshold as a function of grain diameter (d) and temperature. The saltation thresholds are shown for grain compositions of either basalt or $SO_2$, and are also

**Fig. 4 The saltation threshold compared with outgassing velocities.** The lines denote the saltation threshold as a function of temperature. Thick lines are the threshold for $SO_2$ grains, while the thinner lines are for basalt grains. The range of vapor outgassing velocities for the considered temperatures are shown as the shaded box.

compared with the calculated initial outgassing velocities. When the outgassing velocity is higher than the saltation threshold for a given grain size, saltation may occur. At 155 K, saltation becomes possible (in the immediate vicinity of the outgassing) for grain diameters between 20 $\mu$m and 1 mm in size, i.e. for all $SO_2$ grain sizes inferred at Prometheus Patera[25]. Above this temperature, saltation is feasible for all grains between 10 $\mu$m and 1 cm in size. A second minimum is found on each curve where $d > 10^{-3}$ m. This is the point where Kn = 0.01, and is a result of changes to the drag relation used. For grains smaller than this point, noncontinuum draft effects are incorporated via the Cunningham correction, and for larger grains the no-slip condition is used (see Methods).

**Aeolian characteristics of observed linear ridge features.** Given the possibility of saltation at lava-frost margins, we turn to examining observational evidence for aeolian bedforms on Io. Ridges visible in some of the highest resolution images from the Solid State Imager on the _Galileo_ probe (see Fig. 3) have been catalogued[9], with the suggestion that their origin was from tidal forces due in part to the assumption that saltation was impossible[10]. We have demonstrated that saltation may be

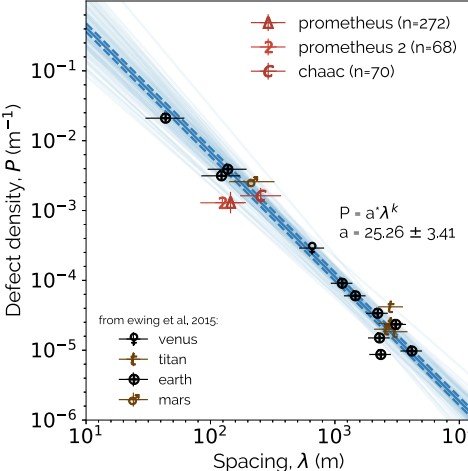

**Fig. 5 Dune defect density vs crest spacing across planetary bodies.** The measurements from this work for Io are compared with the power law trend fit to observations on other planetary bodies[29]. The number of ridges measured at each site on Io are denoted by $n$. The calculation method for the error bars are described in the "Ridge crest digitization" and "Best-fit trendline for defect density to crest spacing ratio" sections of Methods. Our maximum likelihood fit to the data from other bodies (see Methods) is shown as the thick, dark blue line, with the intrinsic $1\sigma$ scatter of that trend indicated by the dashed lines. The best fit parameters are shown. The thin, transparent lines are a sample of less likely fits. Posterior probability distributions for the fit parameters are shown in Supplementary Fig. 5.

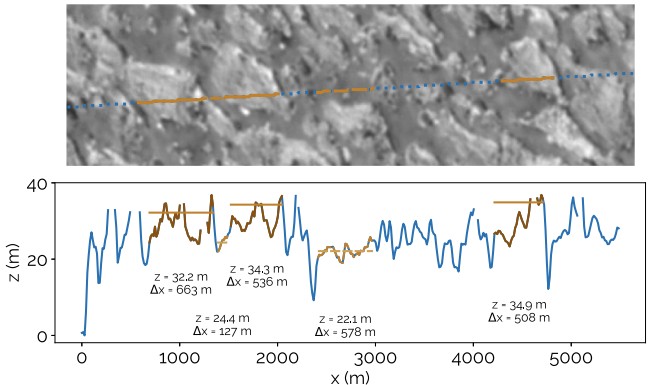

**Fig. 6 Estimated heights of the Chaac Patera ridges from photoclinometry.** The top panel is from the same source image from near Chaac Patera that is shown in Fig. 3c, but is here unprojected. The line is the trace of a sun ray, with solid brown regions corresponding to ridge pixels and dashed brown regions corresponding to inter-ridge pixels, both of which are used to estimate heights. The lower panel shows the heights derived with photoclinometry corresponding to locations in the upper panel. The horizontal lines show the ridge and inter-ridge heights that we extract (see Methods), which are displayed as the $z$ values. The widths of each region are indicated as $\Delta x$.

possible at lava-frost interaction regions. Furthermore, all of these ridges, with the exception of one site, are located directly adjacent to paterae, precisely where such lava-frost interactions are expected to occur (see Fig. 3b). For these reasons, we consider again an aeolian origin for these ridges.

The most extensive high resolution images, at the scale of ~10 m per pixel, of the ridges are found adjacent to the lava flows at Prometheus Patera (see Fig. 3a, b). We measure the ratio of ridge crest spacing to the crestline defect density, which is the density of breaks in the crestlines[28]. For dunes, a power law trend in this ratio is observed across planetary bodies[29]. This trend is thought to arise from the self-organization process of smaller dunes merging together to increase the crest spacing while simultaneously decreasing the defect density[30]. We find that both measured sites at Prometheus Patera fall within $2\sigma$ from the best fit trend calculated for other planetary bodies, consistent with dune-field patterns (see Fig. 5; see Methods).

Observations of ridges at Chaac Patera consist primarily of a single high resolution image (see Fig. 3c and 6). These are consistent with the power law trend for dunes on other bodies at the $1\sigma$ level (see Fig. 5). The higher, 85.86° phase angle of this image, vs ~35° at Prometheus, makes possible photoclinometry, or the use of a surface phase function to derive surface slopes and heights. A rudimentary analysis assuming a Lambertian surface phase function (see Fig. 6; Methods) indicates that the Chaac ridges are asymmetric, with one steep face and one that is gradually sloped, both perpendicular to the longest axis. This profile bears similarity with transverse dunes. We also find height-to-width ratios of ~0.011–0.025, which are comparable to those measured for terrestrial stabilized transverse dunes[31].

The final image of ridges with resolved crestline defects is at Hi'iaka Mons, and is the one site not adjacent to a patera. We agree with the assessment that their presence on a local topographic high and a crestline sinuosity that is so high so as to interrupt adjacent crestlines, suggest an independent origin,

and that these are unlikely to be aeolian features (see Supplementary Fig. 7)[10].

## Discussion

The consistency of these patterns with other dune fields around the solar system highlight the robustness of the self-organization of bedform patterns in a wide range of boundary conditions. Nevertheless, differences between the ridge patterns on Io and other dune fields exist that could arise from Io's unique boundary conditions or other wind-driven pattern-forming surface processes. Breaks in the crestline patterns at Prometheus Patera could be interpreted as isolated features rather than part of the overall dune pattern. The deviation of the Prometheus Patera features from the power law trend for defect density-to-crest spacing manifests in terms of narrowly spaced crests for their defect density. This may signal changes or disruption to the pattern-forming process[29]. Height-to-width measurements are not currently feasible for the Prometheus ridges, but making those measurements would allow for a more robust assessment of their being dunes compared to pattern characteristics alone. Future measurements of the heights of these features may confirm their dune-like characteristics, or their similarities to other features such as erosional yardangs or to penitentes, linear features formed directly by sublimation.

At Chaac Patera, the albedo differences between the ridges and inter-ridge regions may be indicative of material differences. Where this occurs, the ridges may be partially buried signaling a depositional volcanic, atmospheric, or aeolian process.

With regards to the tidal formation hypothesis, rough alignment of the ridges with the tidal stress field was found, however deviations do exist and there does not currently exist an explanation for why the ridges orient perpendicular to north-south stresses in some situations and to east-west stresses in others, when both stresses are present at a given location[10]. Furthermore, on Europa, where tidal stresses are known to manifest as surface features, features such as double ridges and cycloids do not form with a consistent wavelength over the span of multiple features[32], as is observed for the Ionian ridges.

We consider future observations to be necessary to elucidate the origins of these features, but suggest that the existence of a

mechanism for saltation on Io, as well as certain ridge characteristics, justify consideration of an aeolian origin.

From predominantly analytical arguments, we have suggested that saltation on Io may be possible. Nevertheless, the lava-frost interactions are dynamic, under the already time-variable Ionian atmosphere. Future studies would benefit from examining the time evolution of these interactions. Furthermore, consideration of secondary effects on particle saltation may be fruitful. Electrostatic processes—such as fracto and triboelectrification—have been implicated in the dynamics of granular materials on Earth, Mars, and Titan[33,34] and may also modulate transport of sediments on Io. The effects of electrostatics may be exacerbated on Io given the exposure of grains to an ambient plasma and to strong electric and magnetic fields[35].

Even if restricted to the margins of lava flows, the occurrence of saltation on Io would suggest surface modification mechanisms more complicated than previously considered. Furthermore, we restricted our analysis to the lava-frost margins, but there may exist as much as 10–20% of the equatorial plains with elevated-above-ambient temperatures of 110–200 K[25]. If a mechanism exists whereby the ambient pressure of this frost can also be raised, saltation could be occurring over a much broader scale. Our mechanism for saltation, combined with the aeolian-like features of the Ionian ridges, suggest that Io may be among other planetary bodies with tenuous and transitory atmospheres, including Pluto[36] and Comet 67P[22], where aeolian sediment transport is occurring.

## Methods

**Turbulent vapor flow**. To characterize the near-surface flow characteristics of Io's atmosphere, we compute the Reynolds number (Re). The Reynolds number is a non-dimensional number that is used to assess whether a flow is laminar or turbulent. We consider those flows with $\mathrm{Re} > 10^3$ to be transitionally turbulent.

$$\mathrm{Re} = \frac{\rho u L}{\mu} = \frac{\rho u \delta_i}{\mu} \tag{1}$$

where $\rho$ is the fluid density (in this case the atmosphere or the outgassing vapor), $u$ is the flow speed, $\mu$ is the dynamic viscosity, and $L$ is a characteristic linear dimension. $L$ is chosen based on the largest relevant turbulent scales. In our case, we set $L = \delta_i$, which is the height of the viscous sublayer of the atmospheric boundary layer.

The height of $\delta_i$ on Io is not known, and furthermore would vary as a function of the flow types we consider here (the atmosphere vs outgassing vapor flows). We therefore model it as a free parameter, and we report $\mathrm{Re}(\delta_i)$. Nevertheless, we bound the value as being between the $\delta_i$ of ~ mm scale on Earth to the ~1 m predicted for Comet 67P[22].

For the flow velocity, we use 300 m/s for the ambient atmosphere as has been measured[7], and use the velocity calculated in the "Outgassing velocities" section for vapor flows originating from SO$_2$ sublimating at above ambient temperatures and pressures.

Computed Reynolds numbers are shown in Fig. 1.

**Sublimation vapor pressure**. An accurate calculation of the vapor densities resulting from SO$_2$ sublimation is necessary, as these densities are used to assess whether turbulent flow will occur and whether saltation is possible. We employ a polynomial extrapolation relation[37], which uses the Clausius-Clapeyron relation in concert with the internal partition function of SO$_2$ to account for the temperature dependence of the sublimation enthalpy ($\Delta H_{sub}$). The temperature dependence of $\Delta H_{sub,SO2}$ is not reported in the literature. In turn, this method utilizes experimental measurements of the temperature dependence of the solid heat capacity ($C_{P,sol}$) at low temperatures. This polynomial extrapolation is found to reproduce available experimental measurements, between 120 K and 197.64 K (the triple point), with an accuracy of ±9.6%. For the comparison of this extrapolation to available data, the reader is referred to Fig. 26 and section 3.2.38 of Fray et al. 2009[37]. The polynomial is of form:

$$ln(P_{sub}) = A_0 + \sum_{i=1}^{n} \frac{A_i}{T^i} \tag{2}$$

The boiling curve shown in Fig. 2b is evaluated using the empirical Antoine equation from the NIST Database[38]. The melting curve is an approximation shown only for reference, as it assumes an enthalpy of fusion that is constant with temperature (see Supplementary Table 1 for adopted values and references).

**Outgassing velocities**. To calculate the outgassing velocity of a sublimating volatile capped by a porous surface layer, we utilize the relation derived for sublimating water ice on Comet 67P[22]. This relation utilizes the kinetic theory of gases and assumes that a sublimated gas molecule will collide with grains in the porous layer, but not with other molecules. Additional assumptions include a half Maxwell-Boltzmann velocity distribution for the gas molecules, and perfect absorption of the gas molecules when returning to the ice.

The outgassing velocity is then calculated by numerically solving the following energy flux balance for the velocity ratio $\Upsilon_o$:

$$\frac{1}{2}\Upsilon_0(\Upsilon_0^2 + \pi)\rho_0(V_{th}^0)^3 = \frac{7\pi}{16}p_c\left[\frac{1}{4}F\rho_{sat}(V_{th}^i)^3 - q_-(V_{th}^0)^2\right] \tag{3}$$

where $\Upsilon_0 \equiv u_0/V_{th}^0$ is the ratio of the outgassing velocity to the thermal velocity of the vapor at the surface. The solution of $\Upsilon_0$ can thus be used to calculate the outgassing velocity $u_0$. The thermal velocity of the vapor at the surface is $V_{th}^i = \sqrt{8k_B T_i/(\pi m)}$, where $k_B$ is the Boltzmann constant, $T_i$ is the temperature of the sublimating ice, and $m$ is the molecular mass of the gas. $F$ is the ice surface fraction (for which the ultimate solution using Eqs. (3), (6) and (7) shows no dependence) and $\rho_{sat}$ is the saturation vapor density at $T_i$. $q_-$ is the vapor mass flux of molecules entering the porous layer from the atmosphere:

$$q_- = \frac{1}{4}f(\Upsilon_0)\rho_o V_{th}^0 \tag{4}$$

where $f(\Upsilon_0)$ is defined as the function:

$$f(\Upsilon_0) \equiv e^{-4\Upsilon_0^2/\pi} - 2\Upsilon_0\left[1 - \mathrm{erf}\left(\frac{2\Upsilon_0}{\sqrt{\pi}}\right)\right] \tag{5}$$

The expressions for $\rho_0$ and $V_{th}^0$ to be substituted into Eqs. (3) are obtained from mass and momentum flux balances respectively:

$$\rho_0 = F p_c \rho_{sat} \frac{\pi[f(\Upsilon_0)(p_c - 2) + 2] + 16\Upsilon_0^2}{\pi[f(\Upsilon_0)p_c + 4\Upsilon_0]^2} \tag{6}$$

$$V_{th}^0 = V_{th}^i \frac{\pi[f(\Upsilon_0)p_c + 4\Upsilon_0]}{\pi[f(\Upsilon_0)(p_c - 2) + 2] + 16\Upsilon_0^2} \tag{7}$$

Lastly, $p_c$ is the probability for a molecule to cross the porous layer, and is proportional to the ratio of the diameter of the grains $d$ making up the granular layer to the height of the granular layer $h$, which is effectively a convolution of porosity and the height of layer. I.e. $p_c \propto d/h$.

For the derivations of the above relations, the reader is referred to the "Porous Subsurface Layer" section of the Supporting Information for Jia et al. 2017, as well as equations S42 – S51 in that work[22].

**Saltation**. In assessing the initial coupling of grains to the gas flow, we calculate the Stokes number (Stk). The Stokes number is a nondimensional parameter that indicates the coupling of suspended particles to their carrier fluid. For $\mathrm{Stk} \gg 1$, particles are poorly coupled to their carrier fluid and largely follow ballistic trajectories.

$$\mathrm{Stk} = \frac{\tau u}{d} \tag{8}$$

where $d$ is the grain diameter, $u$ is the flow velocity, and $\tau$ is the reaction time of the grains to the fluid. We adopt for $\tau$:

$$\tau = \begin{cases} \rho_s \frac{d^2}{18\mu} & \text{if } \mathrm{Kn} < 0.01 \\ \rho_s \frac{Cd^2}{18\mu} & \text{if } \mathrm{Kn} > 0.01 \end{cases} \tag{9}$$

where the Knudsen number Kn is evaluated using Eq. (10) and the Cunningham correction $C$ using Eq. (18). $\rho_s$ is the density of the solid grains, and $\mu$ is the dynamic viscosity of the fluid.

A large portion of the vapor density and grain size parameter space under consideration is in the high Knudsen number ($Kn$) regime:

$$\mathrm{Kn} = \frac{\ell}{d} \tag{10}$$

where the mean free path ($\ell$) is comparable to or larger than the grain diameter ($d$).

For this reason, a saltation framework is required that accounts for non-continuum effects on particle drag. We utilize the saltation relations which were derived for use in the dilute gas/high Knudsen number regime for application to Comet 67P[22]. Their expression for the saltation threshold ($u_t$) reads:

$$u_t = \sqrt{\Theta_t(\rho_s/\rho_g - 1)gd} \tag{11}$$

where $\rho_g$ is the gas density and $g$ is the gravitational acceleration. The cohesion-adjusted-threshold-Shields-number ($\Theta_t$) is calculated:

$$\Theta_t = \Theta_t^0\left[1 + \frac{3}{2}\left(\frac{d_m}{d}\right)^{5/3}\right] \tag{12}$$

where $d_m$ is the typical grain diameter below which cohesive effects become important. The factors by which $\Theta_t$ are multiplied originate from studies of the

adhesion forces between grain surfaces that are rough at the nanometer scale, specifically section 2 of Bocquet et al. 2002[39]. This is expected to be the case for any micron-scale basalt on Io which would have to be generated through fragmentation. In the case of $SO_2$ grains, frequent diurnal cycling between deposition and sublimation at the grain surfaces may lead to considerable surface roughness. This is the case with scalloping and hexagonal pits that are seen with electron microscopes for water ice undergoing the same process[40], although this has not been tested for $SO_2$ ice. $d_m$ is calculated:

$$d_m \simeq (9.81/1.796)^{(2/5)} \times 10\,\mu m \simeq 19.7\,\mu m \tag{13}$$

where 10 μm is the $d_m$ value for Earth, 9.81 m/s$^2$ is the gravitational acceleration on Earth, and 1.796 m/s$^2$ is the gravitational acceleration on Io. Calculating the Ionian $d_m$ by scaling the gravitational accelerations of Earth and Io makes the assumption that the same, predominantly van der Waals and capillary, cohesive forces are operating for grains on Io, as is the case on Earth. This is reasonable for basalt, although in the case of $SO_2$, the relative strength of additional interparticle forces such as electrostatics[33] are currently unknown.

The threshold Shields number $\Theta_t^0$ is calculated:

$$\Theta_t^0 = 2\left(\frac{d_\nu}{d}\right)^{3/2} S_t^{1/2} + \frac{S_t \kappa^2}{\ln^2(1+1/2\xi)} \tag{14}$$

where $\xi$ is the hydrodynamic roughness scaled to the grain diameter, for which the value of 1/30 is adopted[22]. The threshold Shields number is the nondimensionalized shear stress above which the probability of fluid entrainment exceeds zero for a grain of diameter $d$[13]. The viscous size ($d_\nu$) is calculated:

$$d_\nu = (\rho_s/\rho_g - 1)^{-1/3} \nu^{2/3} g^{-1/3} \tag{15}$$

where $\nu$ is the dynamic viscosity of the gas. $S_t$ is the threshold value of the flow velocity at the grain scale, and is calculated:

$$S_t = \frac{1}{16}\left[\left(s^2\left(\frac{d_\nu}{d}\right)^{3/2} + 8\left(\frac{2\mu}{3}\right)^{1/2}\right)^{1/2} - s\left(\frac{d_\nu}{d}\right)^{3/4}\right]^4 \tag{16}$$

with $s$ being a parameter that modifies the drag term depending on the Knudsen number as defined in Eq. (10). We choose to define a cutoff value of Kn = 0.01:

$$s = \begin{cases} 5 & \text{if } Kn < 0.01 \\ \sqrt{25/C} & \text{if } Kn > 0.01 \end{cases} \tag{17}$$

with $C$ being the Cunningham correction:

$$C = 1 + \frac{2\ell}{d}(1.257 + 0.4\,\exp(-0.55d/\ell)) \tag{18}$$

As discussed in the "Lava-frost interactions and sublimation vapor pressures" section of the main text, grain scale turbulence is a pre-condition for saltation. To ensure that the initiation of motion predicted using the saltation threshold $u_t$ will be followed by saltation and not particle creep, we only apply the above relations for conditions in which we've determined $Re(\delta_i) > 10^3$, indicating that turbulent flow is present. Furthermore, the adopted saltation relation does include dependence on Re, specifically the term $d_\nu$ can be rearranged as:

$$d_\nu = \frac{(\rho_s - \rho_g)^{3/2}}{\rho_g g^{1/2} Re_*^{3/2} u^{3/2} \mu^{1/2}} \tag{19}$$

where $Re_*$ is the particle Reynolds number, i.e. for which the length scale $L$ used in its calculation is the particle diameter $d$.

For derivations of the above relations, the reader is referred to the "Transport Threshold" section of the Supporting Information for Jia et al. 2017 and equations S57 – S64 in that work[22,41].

**Ridge crest digitization**. The images and measurements of ridge properties utilize the highest resolution imagery of those features from the Solid State Imager on the *Galileo* probe. The crestlines of the ridges were digitized manually within the largest boundary rectangle that could be drawn within the image, while still parallel to the crestlines (see Supplementary Fig. 4). The digitizations were used to measure crest spacing and crestline defect density (plotted in Fig. 5 and reported in Supplementary Table 2). This was done using the QGIS software at scales ranging from 1:7000 to 1:11000. The geographical coordinates of the mapped regions as well as the image ID's are tabulated in Supplementary Table 2.

After the crestlines were digitized, the perpendicular distance between each adjacent crest was digitized as a measurement of the crestline spacing[42] (see Supplementary Fig. 4). This generates a statistcal sample of the crestline spacings within a given field of ridges. Because the histogram of crest spacings within a ridge field was often non-Gaussian, for the measurements shown in Fig. 5 and tabulated in Supplementary Table 2, the reported value is the median of all crest spacings, while the uncertainties are the ± 34.1 percentiles.

The defect density is calculated as an aggregate statistic for the entire ridge field[28], wherein the number of defects within the rectangular grid are divided by the total length of the crestlines within that field. Because this method only results

in the calculation of a single defect density value per field, to quantify the uncertainty in this measurement, two additional mappings were conducted for each ridge field. The main source of the uncertainty in the defect density is ambiguity stemming from the resolution of the imagery over whether two adjacent ridges are continuous or not (if they are, there is no defect. If they are not, there is a defect). In the second mapping, all ridges over which there is ambiguity about their linkage are considered to have an interjoining defect (i.e. this will result in a maximum number of defects, and in turn a maximum defect density). In the third mapping, all ridges with ambiguity are considered to be a single continuous feature (i.e. minimum defect density). The difference between these maximum and minimum values from the original mapping are displayed in Fig. 5 and reported in Supplementary Table 2 as the uncertainty in the defect density measurement.

**Best-fit trendline for defect density to crest spacing ratio**. We generate a best-fit trendline for the defect density to crest spacing ratio on other planetary bodies, with which to compare the Ionian ridges. The data that are fit to are exclusively from Ewing et al. 2015 and works quoted therein. We use uncertainties on the crestline spacing from references cited within that work, when available. Because uncertainties were not calculated for the crestline spacings at sites on Titan and Venus, both of which were observed using Synthetic Aperture Radar (SAR), we use as a proxy the uncertainty calculated for a mapping of the crestline spacings of the Belet dune field on Titan as measured using Cassini SAR data[43]. We are not able to compare with recent data measuring crestline interaction densities[30], as these cannot be directly compared to defect densities, and are more difficult to discern in lower resolution images.

For this fit, we use the maximum likelihood method, in which we minimize the orthogonal distance between each data point and a best-fit line in log-log space. Because the scatter around the trend has some physical origin in the form of the level of environmental degradation of dune fields that are no longer active[29], in our fit we include a parameter for intrinsic orthogonal scatter of the line. The ultimate fit is expressed as a power law in linear space and shown in Fig. 4.

To generate the fit, we maximize the log likelihood function[44]:

$$\mathcal{L} = K - \sum_{i=1}^{N}\frac{1}{2}(\Sigma_i^2 + V) - \sum_{i=1}^{N}\frac{\Delta_i^2}{2[\Sigma_i^2 + V]} \tag{20}$$

where $V$ is the variance of the allowed intrinsic scatter. The uncertainty Gaussian $\Sigma_i^2$ is:

$$\Sigma_i^2 = \hat{v}^\top S_i \hat{v} \tag{21}$$

and the orthogonal displacement of each data point $\Delta_i$ is:

$$\Delta_i = \hat{v}^\top Z_i - b\cos\theta \tag{22}$$

$\hat{v}$ is a unit vector defined to be orthogonal to the fit line with slope $m$:

$$\hat{v} = \frac{1}{\sqrt{1+m^2}}\begin{bmatrix} -m \\ 1 \end{bmatrix} = \begin{bmatrix} -\sin\theta \\ \cos\theta \end{bmatrix} \tag{23}$$

with $\theta = \tan^{-1}(m)$ defined as the angle made between the fit line and the $x$ axis. $Z_i$ is the column vector for an individual measurement ($x_i, y_i$):

$$Z_i = \begin{bmatrix} x_i \\ y_i \end{bmatrix} \tag{24}$$

and $S_i$ is the covariance tensor, for the uncertainties associated with the measurement ($x_i, y_i$). Note that in our case, we have not determined the covariances between defect density and crest spacing so $\sigma_{xyi} = 0$:

$$S_i = \begin{bmatrix} \sigma_{xi}^2 & \sigma_{xyi} \\ \sigma_{xyi} & \sigma_{yi}^2 \end{bmatrix} \tag{25}$$

We consider the maximum likelihood value to be the best fit line (shown as the bolded line in Fig. 5, with the 1σ intrinsic scatter denoted by the dashed lines). We perform the maximization of the likelihood with respect to the parameter $b_\perp \equiv b\cos\theta$, the orthogonal distance of the line from the origin, $\theta$ and $V$. We then sample the posterior probability distributions for each of the fit parameters using the *emcee* Markov-Chain Monte Carlo algorithm[45]. We find unimodal distributions for all fit parameters, with some covariance between the intercept $b_\perp$ and $\theta$ (see Supplementary Fig. 5).

**Photoclinometric measurements of ridge heights**. Assuming a phase function for the scattering behavior of the surfaces in an image, the reflectances measured by a detector can be related to the average slope of the facets in a pixel. We make the simplistic assumption that all surfaces in the image are Lambertian scatterers. Given this, the average height for a given pixel $m$ in the image, $z_m$, can be calculated as:

$$z_m = \Delta x \tan\left(i - \cos^{-1}\left(\frac{(a_m - a_s)\cos i}{a_0}\right)\right) \tag{26}$$

where $\Delta x$ is the pixel resolution for the image (e.g. in m/pixel), $i$ is the incidence

angle of the Sun and $a_m$ is the albedo or reflectance of the pixel of interest. $a_s$ is the minimum albedo in the image that corresponds to a directly illuminated pixel. $a_0$ is the albedo that is set to correspond to a flat facet. Note that all derived heights, $z$, for the image are with respect to the other pixels in the image, and an absolute height with respect to Io's shape model cannot be derived from photoclinometry.

For our Chaac Patera image, whose photoclinometrically derived heights are shown in Fig. 6, we set $a_0 = 0.25$ which is a value representative of the darker inter-ridge areas. Although the scattering behavior of the surface is unlikely to be truly Lambertian, as is hinted at by the significant scatter in the extracted heights, we do expect a correlation between terrain brightness and height which can be interpreted to first order. This is because the brightest portions of the ridges consistently occur at the crests of the features, sunward of observed shadows (see reconstruction of the observation geometry in Supplementary Fig. 9). If the ridges were exhibiting predominantly specular behavior, one would expect bright facets appearing more randomly throughout the image.

In deriving the height for a given ridge and average inter-ridge heights, as shown in Fig. 6, we calculate the ridge height as the height of the 84.1 percentile pixel (equivalent to $1\sigma$ above the mean for a Gaussian distribution) occupying a given ridge. For the inter-ridge heights, we take the average of the pixels corresponding to individual inter-ridge regions. This methodology ensures that heights used in the analysis are not defined by single pixels with outlier reflectance values.

## Data availability

All data needed to evaluate the conclusion are present in the paper, Supplementary materials, the Source Data file provided with this paper, or through NASA's Planetary Data System (PDS). The data presented in Figs. 1, 2a, 4 and 6, are provided in the Source Data file, with the independent or dependent variable name indicated in the file name. The Ionian ridge pattern measurements in Fig. 5 are tabulated in Supplementary Table 1 in the Supplementary Information. The ridge images taken by the Solid State Imager on the *Galileo* probe were accessed via the Rings Node of the PDS. The PDS query as well as a list of the retrieved images are included in the Source Data file. Source data are provided with this paper.

## Code availability

The Python functions encoding the equations presented in the Methods section are available on GitHub (https://github.com/gdmcdonald1/io_saltation).

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

## Acknowledgements

We thank David Goldstein at The University of Texas at Austin, William McDoniel at Sandia National Laboratories, Alex Hayes at Cornell University, Kirby Runyon at The Johns Hopkins University Applied Physics Laboratory, and Bruno Andreotti at the University of Paris-Diderot for helpful discussions. This work was funded by a startup grant to L.O. by Rutgers University.

## Author contributions

G.D.M. conceived of the project, carried out all reported calculations, and wrote the manuscript. J.S.M.H. assisted with calculations, provided input on the assumptions made in the outgassing velocity calculations and the lava-frost interactions, and generated Fig. 2b. L.O. provided input on the lava-frost interaction geometry and on analysis techniques for the imagery. P.C. provided the technique for and assisted with the estimation of ridge heights and topographic profiles using photoclinometry. J.D. assisted in the use of nondimensional parameters to characterize fluid behavior. R.C.E. and L.K. provided input on the comparisons of the Ionian ridges with different aeolian bedforms.

## Competing interests

The authors declare no competing interest.
