## [Peer Review File · Nature Communications]

REVIEWER COMMENTS

Reviewer #1 (Remarks to the Author):

This is a harmless paper that explores in quantitative detail the previously-proposed hypothesis that SO₂ volatilization might drive gas flows which could transport surface particulates on Io. The suggestion that aeolian processes driven by SO₂ sublimation may have formed the ridges was advanced in Lorenz and Zimbelman 'Dune Worlds: How Wind-blown Sand Shapes Planetary Landscapes', Springer 2014. p.178-180 of that work shows exactly the same Prometheus ridges as in the present paper fig.4b , and suggests they may be aeolian, and documents their wavelength at 100-200m. So the core thesis of this paper is hardly novel, and at the very least, that book should be referenced here. That said, the present paper explores the relevant phenomena with some useful analysis, so it still has incremental value.

Fig 3 – this looks puzzling to me. Most saltation threshold curves are smooth with a single minimum, but there is a 'waviness' to some of the curves here suggestive of a sort of suppressed second minimum around 1E-3m. Is this some sort of curve-fitting artifact of the plotting software? Or If this is real, it should be explained.

Fig 6 – this does not look to me like a uniform-composition area with tilted facets, but rather more like slabs of one material partly buried in another. If so, then the uniform-optical-property assumption behind photoclinometry does not hold.

A striking omission from the paper is the prospect that strong electric fields could modify aeolian transport (noting the high Jovian trapped particle radiation at Io that can even cause aurorae, and can also deposit charge into surface materials). These same authors have noted the possibility of triboelectric charging and the modification of saltation threshold on Titan by electrical effects – the interesting possibilities here surely deserve a mention in this paper too.

Reviewer #2 (Remarks to the Author):

Review by Claire Newman [claire@aeolisresearch.com] - please note that I am happy to clarify any of my comments via email if that would be helpful.

This is a very interesting paper that examines whether flows due to the interaction between lava and SO₂ frost could produce potentially aeolian features (ridges) seen on the surface of Io, rather than the weaker sublimation-condensation flows associated with Io's atmosphere freezing out regularly. The paper provides theoretical estimates of the outgassing velocities and compares these to estimates of the threshold for saltation to occur. It also notes that most ridges are directly adjacent to volcanic craters, and examines the ridges for evidence of an aeolian nature. I highly recommend this paper for publication in the near future - HOWEVER, I also have several major comments that I feel must be addressed first.

My primary concerns are that:

- The paper is laid out somewhat oddly, with early emphasis on providing results over context and background in the introduction.
- Figure 1 is not well-explained and may confuse more than it helps.
- There is insufficient explanation of where the saltation thresholds come from and the conceptual differences between flows and thresholds under different situations.
- Non-dimensional numbers are thrown in too often without providing enough context as to their meaning.
- The references to yardangs seem to be red herrings.

MAJOR COMMENTS:

(1) Lines 42-47:

(a) Odd phrasing here. Suggest e.g. "either dismissal of this possibility due to the anticipated low gas densities and hence wind stresses available in Io's mean sublimation atmosphere"

(b) But more importantly, what is meant by the "mean sublimation atmosphere" needs to be defined.

(c) In fact, I think (b) is part of a deeper issue, in that nowhere in the text is the sublimation-condensation flow on Io described. I assume that by "mean sublimation atmosphere" the authors are referring to the flows associated with 42-hour periodicity of Io moving in and out of Jupiter's shadow, causing the SO₂ frost to freeze and sublime, creating air flow?

OVERALL: Given that this flow is not widely described - and that it has previously been suggested to potentially be responsible for aeolian features, whereas this paper present an alternate hypothesis - I think spending at least a sentence or two on describing what it consists of is needed to set the scene.

(2) The final paragraph of the introduction:

(a) This paragraph attempts to summarize both the methods and the key results. The latter especially belong in the results/discussion sections and the abstract, not the introduction. Removing the results from this paragraph, and making it more of a short 'roadmap' for the rest of the paper, would make valuable room for additional text.

(b) The summary here is also a bit unclear. So some are dunes but others may be yardangs, but they all have density to crest spacing ratios consistent with dunes? Is such a spacing also consistent with yardangs, then? Again, this is probably best left to later sections.

(c) The final sentence of the introduction again feels like a reprise of the abstract and the text seems more suited to the discussion section, not the introduction.

(3) Figure 1 is quite hard to puzzle out and I think could be described and presented far more clearly.

Plot (a) is very small and it's not immediately clear that the x-axis label is shared with plot b due to the placement of the other labels below it. The caption for (a) only talks about Re being shown as a function of δ , but should also mention ρ . ρ is never defined in the caption. I infer it's the outflowing vapor density, but this should be stated more carefully. In addition, I feel that words rather than symbols should be used in the x- and y-axis labels.

Overall, I suggest clearly showing (in the figure) and stating (in the caption) what each plot contains.

(4) The bigger issue, though, is that the key results of Figure 1 are not well explained in the text. Figure 1 is very complicated and consists of multiple plots but is only referred to in passing early in section 2.1 and with none of the explanation that would be needed for anyone but an expert dynamicist to interpret the meaning. I think this could be improved by setting the scene a little for why the Reynolds number is important and explaining what the critical Reynolds number denotes, either here or in the Figure caption (see comment 1).

As an aside, this is why I dislike the placing of a 'Methods' section at the end of an article when the information contained therein is typically required to make sense of what's written earlier. [I really need to stop agreeing to review articles in Nature because this is nearly always an issue. At the same time, authors who choose to submit to Nature should consider how to summarize their methods better in the main text, even if the details must be placed at the end.]

(5) First paragraph of section 2.1:

(a) Perhaps unintentionally, this paragraph has the effect of implying that laminar flows cannot produce saltation. That's clearly incorrect - strong winds do not need to be turbulent to raise particles into saltation on Earth, for example - so the authors are really saying something different here, relating to the various types of thresholds for grain motion that exist (and which are a major topic of research in planetary aeolian studies at present, making this an admittedly tricky thing to delve into). However, I think this needs clarification. See also my previous comment.

(b) See also the sentence beginning on line 195: Again, it is really crucial to (earlier) explain the significance of turbulent flow for initiating saltation in these conditions.

(6) General focus of the paper should be more clearly on the new hypothesis:

(a) The key idea - that the aeolian processes occur due to lava interactions occurring *beneath* a 'snow pack' - is almost buried. Not until Line 113 is this described clearly. Effectively, all of this happening under a sort of 'dome' of SO₂, hence the pressures under discussion are not the pressure of the exposed-to-space atmosphere, but rather the pressure in a 'cave' of sorts. I think this needs to be made clear in the abstract and introduction.

(b) Figure 2b is good at showing this, but the concept shown here could be made clearer in the abstract and overall in the text. This would help the reader follow the key arguments being made.

(c) I also wonder if this paper would be clearer if Figure 1 were moved to the Methods section [given that so little explanation is provided of it in the main text anyway] with only the key points summarized in the main text.

(7) Lines 143-146: This is very jargonistic and hard to follow. A little more explanation would be good.

(8) Lin 147: "Nevertheless..."

This is a bit of a non-sequitur... First the paper is talking about how volatile outgassing flows are complicated, then it's talking about how there's granular material present [presumably on Io, not comets, though this isn't stated]. How are these topics linked via "Nevertheless"?

And then line 152 is back to outgassing flows. Again, the flow [of this paragraph] is a bit strange here.

The sentence starting on line 161 may be the connection between flow speed and granular material, but this certainly wasn't clear earlier.

OVERALL: These sentences could be revised/rearranged for clarity.

(9) Line 172: Compressibility was - I thought - not one of the complications mentioned before. Or was this implied by the mention of difficulties in applying non-dimensionality?

Again, there is a great deal of jargon here.

(10) Line 177: "Thus the direction of the vapor flow will be influenced by the motion of the lava, and favored to flow away from the lava source (see Fig. 2b)."

(a) It's a bit unclear what is meant by this. Figure 2b seems to show a multitude of directions; are you simply saying that the plumes are less likely to point back in the direction of the lava because the flow originates at the lava-frost interface? Hence the only way there can be a component of the flow back towards the lava is if it's forced that direction by its escape route through cracks in the solid SO₂?

(b) OR: Figure 2 marks one flow (the only one appearing to have a component back toward the lava) as "SO2 vapor" and the rest as 'momentum driven plumes'. So is the former meant to have a different formation mechanism? Or is this just accidental?

(11) Line 90 etc. talk about "volcanic plumes" as a source of SO2 frost. Are these the same "plumes" labeled in Figure 2b, which I believe are referred to as "SO2 vapor flows" or "outgassing" in the main text? Or are the latter a different phenomenon? I just want to make sure there are consistent definitions throughout.

(12) Line 188: This again jumps the reader straight into some range of a non-dimensional number for which we have no sense of its importance. Please explain the relevance of the Stokes number *here in the main text*, rather than simply giving numbers.

(13) Line 208: "Fig. 3 shows the saltation threshold as a function of grain diameter and temperature."

(a) WHAT saltation threshold? This is introduced with zero background or references.

(b) Line 420 (buried in the Methods section) finally states "We utilize the saltation relations of Jia et al. 2017" - but this really needs to be referenced in the main text and more explanation is needed than "were derived with use in the dilute gas/high Knudsen number regime in mind."

(c) Lines 462-465: "For the derivations of the above relations as well as additional context, the reader is referred to the "Transport Threshold" section of the Supporting Information for Jia et al. 2017 and equations S57 - S64 in that work."

Referring to the supporting Information from another paper in your methods section is about three levels down from the explanation needed here.

OVERALL: Rather than summarizing the results in the introduction section, I think this paper would benefit from including a 'primer' there about saltation thresholds under different types of conditions. The idea is not to explain everything or derive any equations, but simply to provide the reader with a clearer *conceptual* understanding of how the various situations differ.

(14) All mention of yardangs (including the abstract).

(a) Line 261: "one steep face and one that is gradually sloped"

This immediately made me think of transverse dunes, not yardangs.

(b) Also - as I noted in comment 2b - would you expect yardangs to have a similar density to height ratio as dunes?

(c) My impression is that yardangs are rather symmetric in the direction parallel to winds - i.e., parallel to the ridges. So if the 'steep' and 'gently' slopes here correspond to either side of the ridge line, is that really consistent with yardangs? Isn't Pelletier talking more about the slopes perpendicular to the wind? But perhaps I'm not clear on what you're talking about here.

(d) Lines 281-283: "At Chaac Patera, the steep face is parallel to, not perpendicular to the longest axis, as occurs with yardangs."

Okay - yes, this is what my previous comment (c) is getting at.

OVERALL: I feel like mentioning yardangs is a bit of a red herring.

(15) Methods section:

Too much in this section goes unexplained in the main text, or even here. Even if the gory details need to go here, it is *crucial* that the reader be able to follow the basic concepts without

needing to refer to this section (or seek out the references provided).

**MINOR COMMENTS:
See annotated PDF.**

REVIEWER COMMENTS

Reviewer #1 (Remarks to the Author):

This is a harmless paper that explores in quantitative detail the previously-proposed hypothesis that SO₂ volatilization might drive gas flows which could transport surface particulates on Io. The suggestion that aeolian processes driven by SO₂ sublimation may have formed the ridges was advanced in Lorenz and Zimbelman 'Dune Worlds: How Wind-blown Sand Shapes Planetary Landscapes', Springer 2014. p.178-180 of that work shows exactly the same Prometheus ridges as in the present paper fig.4b , and suggests they may be aeolian, and documents their wavelength at 100-200m. So the core thesis of this paper is hardly novel, and at the very least, that book should be referenced here. That said, the present paper explores the relevant phenomena with some useful analysis, so it still has incremental value.

Fig 3 – this looks puzzling to me. Most saltation threshold curves are smooth with a single minimum, but there is a 'waviness' to some of the curves here suggestive of a sort of suppressed second minimum around 1E-3m. Is this some sort of curve-fitting artifact of the plotting software? Or If this is real, it should be explained.

Fig 6 – this does not look to me like a uniform-composition area with tilted facets, but rather more like slabs of one material partly buried in another. If so, then the uniform-optical-property assumption behind photoclinometry does not hold.

A striking omission from the paper is the prospect that strong electric fields could modify aeolian transport (noting the high Jovian trapped particle radiation at Io that can even cause aurorae, and can also deposit charge into surface materials). These same authors have noted the possibility of triboelectric charging and the modification of saltation threshold on Titan by electrical effects – the interesting possibilities here surely deserve a mention in this paper too.

Reviewer #2 (Remarks to the Author):

Review by Claire Newman [claire@aeolisresearch.com] - please note that I am happy to clarify any of my comments via email if that would be helpful.

This is a very interesting paper that examines whether flows due to the interaction between lava and SO₂ frost could produce potentially aeolian features (ridges) seen on the surface of Io, rather than the weaker sublimation-condensation flows associated with Io's atmosphere freezing out regularly. The paper provides theoretical estimates of the outgassing velocities and compares these to estimates of the threshold for saltation to occur. It also notes that most ridges are directly adjacent to volcanic craters, and examines the ridges for evidence of an aeolian nature. I highly recommend this paper for publication in the near future - HOWEVER, I also have several major comments that I feel must be addressed first.

My primary concerns are that:

- The paper is laid out somewhat oddly, with early emphasis on providing results over context and background in the introduction.
- Figure 1 is not well-explained and may confuse more than it helps.
- There is insufficient explanation of where the saltation thresholds come from and the conceptual differences between flows and thresholds under different situations.
- Non-dimensional numbers are thrown in too often without providing enough context as to their meaning.
- The references to yardangs seem to be red herrings.

MAJOR COMMENTS:

(1) Lines 42-47:

(a) Odd phrasing here. Suggest e.g. "either dismissal of this possibility due to the anticipated low gas densities and hence wind stresses available in Io's mean sublimation atmosphere"

(b) But more importantly, what is meant by the "mean sublimation atmosphere" needs to be defined.

(c) In fact, I think (b) is part of a deeper issue, in that nowhere in the text is the sublimation-condensation flow on Io described. I assume that by "mean sublimation atmosphere" the authors are referring to the flows associated with 42-hour periodicity of Io moving in and out of Jupiter's shadow, causing the SO₂ frost to freeze and sublime, creating air flow?

OVERALL: Given that this flow is not widely described - and that it has previously been suggested to potentially be responsible for aeolian features, whereas this paper present an alternate hypothesis - I think spending at least a sentence or two on describing what it consists of is needed to set the scene.

(2) The final paragraph of the introduction:

(a) This paragraph attempts to summarize both the methods and the key results. The latter especially belong in the results/discussion sections and the abstract, not the introduction. Removing the results from this paragraph, and making it more of a short 'roadmap' for the rest of the paper, would make valuable room for additional text.

(b) The summary here is also a bit unclear. So some are dunes but others may be yardangs, but they all have density to crest spacing ratios consistent with dunes? Is such a spacing also consistent with yardangs, then? Again, this is probably best left to later sections.

(c) The final sentence of the introduction again feels like a reprise of the abstract and the text seems more suited to the discussion section, not the introduction.

(3) Figure 1 is quite hard to puzzle out and I think could be described and presented far more clearly.

Plot (a) is very small and it's not immediately clear that the x-axis label is shared with plot b due to the placement of the other labels below it. The caption for (a) only talks about Re being shown as a function of δ , but should also mention ρ . ρ is never defined in the caption. I infer it's the outflowing vapor density, but this should be stated more carefully. In addition, I feel that words rather than symbols should be used in the x- and y-axis labels.

Overall, I suggest clearly showing (in the figure) and stating (in the caption) what each plot contains.

(4) The bigger issue, though, is that the key results of Figure 1 are not well explained in the text. Figure 1 is very complicated and consists of multiple plots but is only referred to in passing early in section 2.1 and with none of the explanation that would be needed for anyone but an expert dynamicist to interpret the meaning. I think this could be improved by setting the scene a little for why the Reynolds number is important and explaining what the critical Reynolds number denotes, either here or in the Figure caption (see comment 1).

As an aside, this is why I dislike the placing of a 'Methods' section at the end of an article when the information contained therein is typically required to make sense of what's written earlier. [I really need to stop agreeing to review articles in Nature because this is nearly always an issue. At the same time, authors who choose to submit to Nature should consider how to summarize their methods better in the main text, even if the details must be placed at the end.]

(5) First paragraph of section 2.1:

(a) Perhaps unintentionally, this paragraph has the effect of implying that laminar flows cannot produce saltation. That's clearly incorrect - strong winds do not need to be turbulent to raise particles into saltation on Earth, for example - so the authors are really saying something different here, relating to the various types of thresholds for grain motion that exist (and which are a major topic of research in planetary aeolian studies at present, making this an admittedly tricky thing to delve into). However, I think this needs clarification. See also my previous comment.

(b) See also the sentence beginning on line 195: Again, it is really crucial to (earlier) explain the significance of turbulent flow for initiating saltation in these conditions.

(6) General focus of the paper should be more clearly on the new hypothesis:

(a) The key idea - that the aeolian processes occur due to lava interactions occurring *beneath* a 'snow pack' - is almost buried. Not until Line 113 is this described clearly. Effectively, all of this happening under a sort of 'dome' of SO₂, hence the pressures under discussion are not the pressure of the exposed-to-space atmosphere, but rather the pressure in a 'cave' of sorts. I think this needs to be made clear in the abstract and introduction.

(b) Figure 2b is good at showing this, but the concept shown here could be made clearer in the abstract and overall in the text. This would help the reader follow the key arguments being made.

(c) I also wonder if this paper would be clearer if Figure 1 were moved to the Methods section [given that so little explanation is provided of it in the main text anyway] with only the key points summarized in the main text.

(7) Lines 143-146: This is very jargonistic and hard to follow. A little more explanation would be good.

(8) Lin 147: "Nevertheless..."

This is a bit of a non-sequitur... First the paper is talking about how volatile outgassing flows are complicated, then it's talking about how there's granular material present [presumably on Io, not comets, though this isn't stated]. How are these topics linked via "Nevertheless"?

And then line 152 is back to outgassing flows. Again, the flow [of this paragraph] is a bit strange here.

The sentence starting on line 161 may be the connection between flow speed and granular material, but this certainly wasn't clear earlier.

OVERALL: These sentences could be revised/rearranged for clarity.

(9) Line 172: Compressibility was - I thought - not one of the complications mentioned before. Or was this implied by the mention of difficulties in applying non-dimensionality?

Again, there is a great deal of jargon here.

(10) Line 177: "Thus the direction of the vapor flow will be influenced by the motion of the lava, and favored to flow away from the lava source (see Fig. 2b)."

(a) It's a bit unclear what is meant by this. Figure 2b seems to show a multitude of directions; are you simply saying that the plumes are less likely to point back in the direction of the lava because the flow originates at the lava-frost interface? Hence the only way there can be a component of the flow back towards the lava is if it's forced that direction by its escape route through cracks in the solid SO₂?

(b) OR: Figure 2 marks one flow (the only one appearing to have a component back toward the lava) as "SO2 vapor" and the rest as 'momentum driven plumes'. So is the former meant to have a different formation mechanism? Or is this just accidental?

(11) Line 90 etc. talk about "volcanic plumes" as a source of SO2 frost. Are these the same "plumes" labeled in Figure 2b, which I believe are referred to as "SO2 vapor flows" or "outgassing" in the main text? Or are the latter a different phenomenon? I just want to make sure there are consistent definitions throughout.

(12) Line 188: This again jumps the reader straight into some range of a non-dimensional number for which we have no sense of its importance. Please explain the relevance of the Stokes number *here in the main text*, rather than simply giving numbers.

(13) Line 208: "Fig. 3 shows the saltation threshold as a function of grain diameter and temperature."

(a) WHAT saltation threshold? This is introduced with zero background or references.

(b) Line 420 (buried in the Methods section) finally states "We utilize the saltation relations of Jia et al. 2017" - but this really needs to be referenced in the main text and more explanation is needed than "were derived with use in the dilute gas/high Knudsen number regime in mind."

(c) Lines 462-465: "For the derivations of the above relations as well as additional context, the reader is referred to the "Transport Threshold" section of the Supporting Information for Jia et al. 2017 and equations S57 - S64 in that work."

Referring to the supporting Information from another paper in your methods section is about three levels down from the explanation needed here.

OVERALL: Rather than summarizing the results in the introduction section, I think this paper would benefit from including a 'primer' there about saltation thresholds under different types of conditions. The idea is not to explain everything or derive any equations, but simply to provide the reader with a clearer *conceptual* understanding of how the various situations differ.

(14) All mention of yardangs (including the abstract).

(a) Line 261: "one steep face and one that is gradually sloped"

This immediately made me think of transverse dunes, not yardangs.

(b) Also - as I noted in comment 2b - would you expect yardangs to have a similar density to height ratio as dunes?

(c) My impression is that yardangs are rather symmetric in the direction parallel to winds - i.e., parallel to the ridges. So if the 'steep' and 'gently' slopes here correspond to either side of the ridge line, is that really consistent with yardangs? Isn't Pelletier talking more about the slopes perpendicular to the wind? But perhaps I'm not clear on what you're talking about here.

(d) Lines 281-283: "At Chaac Patera, the steep face is parallel to, not perpendicular to the longest axis, as occurs with yardangs."

Okay - yes, this is what my previous comment (c) is getting at.

OVERALL: I feel like mentioning yardangs is a bit of a red herring.

(15) Methods section:

Too much in this section goes unexplained in the main text, or even here. Even if the gory details need to go here, it is *crucial* that the reader be able to follow the basic concepts without

needing to refer to this section (or seek out the references provided).

**MINOR COMMENTS:
See annotated PDF.**

REVIEWER COMMENTS

Reviewer #1 (Remarks to the Author):

The authors appear to have responded adequately to my review

Reviewer #2 (Remarks to the Author):

Second review by Claire Newman.

Thank you for all the changes and the response to my previous comments. I have 5 sets of remaining comments [A-E], although D involves two minor typos and E is a final figure suggestion that can be ignored.

A. My area of expertise is not the onset of saltation. However, I do want to comment briefly on three of the statements in the revised section 2.1:

1. The claim that turbulence "is required" for saltation which is central to this paper's calculations.

I think the main point of arguments about the importance of turbulence is to not rely on using the averaged wind stress [or the wind stress during calm periods] to determine whether saltation is possible. Certainly, when the flow is turbulent, there will be periods when stronger wind stresses exist that may permit lift to occur - provided, of course, that those periods are long enough [as noted by Pähtz et al.]. I feel this is a bit different to saying turbulence is required; what seems to be said here is that stronger winds must exist. But surely, if winds as strong as those involved in a turbulent gust were to exist in more laminar form [i.e. be sustained without variation for longer periods], they would lift even more material.

In other words, by requiring a turbulent flow here, it seems as if you are really making a statement that "we believe that winds are too low in an averaged sense or in periods when there is little gustiness, and that stronger winds experienced during wind gusts are required for lift." Maybe that translates as the same thing in your view - and I'm sure the author list is more expert in this than I am - but it seems confusing to me that this isn't explained more.

Or perhaps it is more the idea that laminar flows can raise particles but that such particles also cease moving very quickly that is really key here? This seems to be what Clark et al. are saying and perhaps it is also a key finding of Pähtz et al. that I'm not grasping. If so, I think more emphasis of and a bit more explanation of this idea is warranted.

2. Statements that laminar flows effectively produce only creep.

The Clark et al. paper is very interesting, basically examining when particles in motion are likely to stop moving. The Pähtz et al. paper is similarly very informative.

The manuscript references Pähtz et al. following the statement that "all laminar flows effectively only produce creep".

However, the response to reviewers references the Clark et al. paper for the statement "while flows that are laminar at our scale of interest possess vertical velocity shear which can theoretically initiate particle motion, any subsequent particle motion manifests in the form of creep."

I can't find a clear statement or a very strong implication of this in either reference, although Clark et al. may come closer.

3. The statement in 2.1 "For saltation, vertical fluid advection from eddies is required to lift particles (Bagnold, 1941; Pähtz et al., 2020)." Is this wording correct? I can see the vertical

component of eddy motions being critical for lifting raised material higher in the boundary layer, but that doesn't seem to be what is meant here. Or perhaps if one extrapolates this to the region right above the surface, you do effectively get a surface lift effect too... Again, this isn't my area, so I must defer to other experts, but I couldn't clearly see this discussed in e.g. Pähtz et al. [although the vertical lift component was, so perhaps it's all in there and I'm just missing it?]

B. Given the emphasis on turbulence being required, is there not then an inconsistency associated with using saltation thresholds that don't include such turbulent effects? Actually, the saltation threshold that is eventually used in the Methods section *does* - according to the text below equation 18 - include some Re dependence. However, it's unclear where this comes in, perhaps because of some confusing definitions and omissions in this section [see also next comment].

C. The Jia et al. saltation threshold.

1. In Methods, Θ_t is first defined as the "threshold Shields number" and the equation for it includes a factor Θ_{t_0} [equation 12]. However, Θ_{t_0} is subsequently *also* defined as the "threshold Shields number" and given in equation 14. This is immediately confusing.

2. There is a mention of the cohesion effect [which I infer is given by the second term on the right hand side (RHS) of equation 12], then the factor in front of the RHS of the equation (whatever it's called) is given in the following equations - but with almost no interpretation of its physical meaning. Is it entirely related to the slip-flow correction?

3. Also - as noted in comment B - while the added text talks about a Reynolds number dependence, the description of the Jia et al. threshold does not make this clear at all.

D. A few minor typos involving new text. The text in 2.2 defining the Stokes number has a typo I think: "suspended" rather than "suspended". Also, in the next paragraphy "their is" should be "there is".

E. Figure 1 is much improved but panel a is still very small. However, if the journal is happy with the print size, that's fine.

2ND REVISION: RESPONSE TO REVIEWERS

We appreciate the time of the reviewers in suggesting the following corrections. We have expanded upon the discussion of why turbulence is necessary for saltation, including a segue from the terrestrial perspective, as well as clarified several terms of the saltation relations, particularly those related to interparticle forces and grain-scale turbulent flow. These changes have improved the clarity of the manuscript, particularly in contextualizing the unique challenges associated with assessing saltation under tenuous atmospheric conditions.

We respond in detail to each comment below, and indicate the relevant line numbers in the document of differences between the initial submission and revision, which we have also submitted.

Thank you,

GD McDonald, J Méndez Harper, L Ojha, P Corlies, J Dufek, RC Ewing, and L Kerber

REVIEWER COMMENTS

Reviewer #1 (Remarks to the Author):

The authors appear to have responded adequately to my review

We thank you for your time in reviewing our 1st revision and are happy to hear that you find our revisions satisfactory.

REVIEWER COMMENTS

Reviewer #2 (Remarks to the Author):

Second review by Claire Newman.

Thank you for all the changes and the response to my previous comments. I have 5 sets of remaining comments [A-E], although D involves two minor typos and E is a final figure suggestion that can be ignored.

A. My area of expertise is not the onset of saltation. However, I do want to comment briefly on three of the statements in the revised section 2.1:

1. The claim that turbulence “is required” for saltation which is central to this paper’s calculations.

I think the main point of arguments about the importance of turbulence is to not rely on using the averaged wind stress [or the wind stress during calm periods] to determine whether saltation is possible. Certainly, when the flow is turbulent, there will be periods when stronger wind stresses exist

that may permit lift to occur - provided, of course, that those periods are long enough [as noted by Pächtz et al.]. I feel this is a bit different to saying turbulence is required; what seems to be said here is that stronger winds must exist. But surely, if winds as strong as those involved in a turbulent gust were to exist in more laminar form [i.e. be sustained without variation for longer periods], they would lift even more material.

In other words, by requiring a turbulent flow here, it seems as if you are really making a statement that “we believe that winds are too low in an averaged sense or in periods when there is little gustiness, and that stronger winds experienced during wind gusts are required for lift.” Maybe that translates as the same thing in your view - and I’m sure the author list is more expert in this than I am - but it seems confusing to me that this isn’t explained more.

Or perhaps it is more the idea that laminar flows can raise particles but that such particles also cease moving very quickly that is really key here? This seems to be what Clark et al. are saying and perhaps it is also a key finding of Pächtz et al. that I’m not grasping. If so, I think more emphasis of and a bit more explanation of this idea is warranted.

We have clarified in the manuscript that we are referring specifically to turbulence “at length scales relevant to the saltation of sand dune grains.” In the Methods section, we specify that the length scale which we use in calculations is the height of the viscous sublayer of the atmospheric boundary layer (line 443). We have also added language introducing the consideration of grain-scale turbulence starting from the terrestrial perspective, where “winds are effectively always turbulent at scales relevant to the saltation of sand dune grains” (while including a full-fledged calculation in the manuscript itself is out of scope, we have included it here for your reference). The final paragraph of your comments is close to the situation we are describing---that laminar flows can “initiate particle motion” but that “transport via saltation will not occur due to the lack of vertical fluid advection.”

Lines 101-117.

Terrestrial calculation:

$$Re \geq \frac{\rho u \delta_i}{\mu} = \frac{(1.23 \text{ kg/m}^3)(2.93 \text{ m/s})(3 \times 10^{-3} \text{ m})}{1.8 \times 10^{-5} \text{ kg/m}\cdot\text{s}} \approx 600$$

The value of 2.93 m/s for u stems from taking the minimum terrestrial saltation threshold friction velocity of $u_* \sim 0.2$ m/s (Iversen and White 1982, Shao and Lu 2000) and using the law of the wall to calculate the freestream velocity u at the viscous sublayer height of $\delta_i = 3 * 10^{-3}$ m. For that calculation, we use a surface roughness z_0 of 7.5 μm , which is 0.1 times the surface roughness element of 75 μm ---the diameter of the grain for which the terrestrial saltation threshold minimum of 0.2 m/s occurs. It is because we have used the minimum terrestrial threshold that the above calculation manifests as a lower limit on the terrestrial Re . For any conditions under which saltation is occurring, the Reynolds number on Earth will be above 600, close to or greater than the $Re = 10^3$ value for transitionally turbulent flows.

2. Statements that laminar flows effectively produce only creep.

The Clark et al. paper is very interesting, basically examining when particles in motion are likely to stop moving. The Pächtz et al. paper is similarly very informative.

The manuscript references Pächtz et al. following the statement that “all laminar flows effectively only produce creep”.

However, the response to reviewers references the Clark et al. paper for the statement “while flows that are laminar at our scale of interest possess vertical velocity shear which can theoretically initiate particle motion, any subsequent particle motion manifests in the form of creep.”

I can't find a clear statement or a very strong implication of this in either reference, although Clark et al. may come closer.

The second half of the latter statement does indeed come from the Pächtz et al. 2020 paper rather than the Clark et al. 2017 paper. We apologize for the confusion and the manuscript now only cites the Pächtz et al. 2020 paper. The first part of the statement is simply reminding the reader that laminar flows can possess vertical velocity shear, with velocity profiles that decay linearly when approaching the wall. Such shear can still initiate particle motion in terms of overcoming the drag force in the force balance calculation typically used for quantifying saltation thresholds (i.e. Fig. 1 of Shao and Lu 2000).

3. The statement in 2.1 “For saltation, vertical fluid advection from eddies is required to lift particles (Bagnold, 1941; Pächtz et al., 2020).” Is this wording correct? I can see the vertical component of eddy motions being critical for lifting raised material higher in the boundary layer, but that doesn't seem to be what is meant here. Or perhaps if one extrapolates this to the region right above the surface, you do effectively get a surface lift effect too... Again, this isn't my area, so I must defer to other experts, but I couldn't clearly see this discussed in e.g. Pächtz et al. [although the vertical lift component was, so perhaps it's all in there and I'm just missing it?]

This point has similarly been clarified by our response to your Comment 1, with our language that laminar flows can “initiate particle motion” but that “transport via saltation will not occur due to the lack of vertical fluid advection.”

Lines 109 – 112.

B. Given the emphasis on turbulence being required, is there not then an inconsistency associated with using saltation thresholds that don't include such turbulent effects? Actually, the saltation threshold that is eventually used in the Methods section *does* - according to the text below equation 18 - include some Re dependence. However, it's unclear where this comes in, perhaps because of some confusing definitions and omissions in this section [see also next comment].

We now reemphasize in the methods section that we use the saltation threshold to test for whether initiation of motion will occur, and then ensure through the use of our Re criteria that the particle can subsequently saltate rather than exclusively creep. We also explicitly show the Re dependence contained in the saltation threshold by rearranging the d_v factor.

Lines 603 – 622.

C. The Jia et al. saltation threshold.

1. In Methods, Θ_{t_0} is first defined as the “threshold Shields number” and the equation for it includes a factor Θ_{t_0} [equation 12]. However, Θ_{t_0} is subsequently *also* defined as the “threshold Shields number” and given in equation 14. This is immediately confusing.

The two terms were indeed previously differentiated with the former being the “threshold Shields number including cohesive effects,” however we can understand that it’s can be ambiguous that the latter part of that language is actually part of the variable name. We have eliminated this ambiguity by now referring to Θ_t as the “cohesion-adjusted-threshold-Shields-number.”

Line 555.

2. There is a mention of the cohesion effect [which I infer is given by the second term on the right hand side (RHS) of equation 12], then the factor in front of the RHS of the equation (whatever it’s called) is given in the following equations - but with almost no interpretation of its physical meaning. Is it entirely related to the slip-flow correction?

We have clarified that the factors correct for the nm scale surface roughness elements of the grains vs. perfectly smooth spheres. As with regards to the origin of the factors---the reviewer’s comments made us aware that we were not sure of the original derivation in the literature. Neither the Jia et al. 2017 manuscript, nor any related manuscripts using the same underlying saltation threshold derivation (Claudin and Andreotti 2006, Andreotti et al. 2021) indicated the origin of the factors. We hope the reviewer will be satisfied in hearing that we reached out to Bruno Andreotti from the above papers, who indicated that the lack of an original citation for the factors was an oversight, and traced the origin of these factors to the paper which we now cite: Bocquet et al. 2002.

Lines 564 – 568.

We also now indicate the physical meaning of the Shields number.

Lines 591 – 594.

3. Also - as noted in comment B - while the added text talks about a Reynolds number dependence, the description of the Jia et al. threshold does not make this clear at all.

See our response to comment B, this is now clarified in:

Lines 603 – 622.

D. A few minor typos involving new text. The text in 2.2 defining the Stokes number has a typo I think: “suspensed” rather than “suspended”. Also, in the next paragraphy “their is” should be “there is”.

Thank you for noticing these, they have now been corrected.

Lines 252, 266.

E. Figure 1 is much improved but panel a is still very small. However, if the journal is happy with the print size, that’s fine.

We have slightly widened panel a), which hopefully aids in its interpretation.

REVIEWERS' COMMENTS

Reviewer #2 (Remarks to the Author):

Thank you for the further changes and explanations. I am satisfied with the responses to my final comments and feel that this very interesting manuscript is ready for publication.

Claire Newman.